# Study of a High-Temperature and High-Density Water-Based Drilling Fluid System Based on Non-sulfonated Plant Polymers

**DOI:** 10.3390/polym14204433

**Published:** 2022-10-20

**Authors:** Jiang Xu, Fu-Chang You, Shu-Sheng Zhou

**Affiliations:** 1Sinopec Research Institute of Petroleum Engineering, Beijing 100101, China; 2Petroleum Engineering College, Yangtze University, Wuhan 430100, China; 3Jiahua Technology Co., Ltd., Jingzhou 434000, China

**Keywords:** filtrate reducer, viscosity reducer, water-based drilling fluid, polymerization reaction, non-sulfonated plant polymer, high temperature

## Abstract

The environment-friendly water-based drilling fluid system developed for the petroleum development industry cannot successfully withstand temperatures up to 180 °C, and most high temperature-resistant additives with sulfonic acid groups that have been successfully applied to water-based drilling fluid are not good for environmental protection. In order to solve the above technical problems, a non-sulfonated filtrate reducer and viscosity reducer with resistance to high temperature were prepared by using humic acid, lignin and a multifunctional monomer as raw materials. In laboratory experiments, the molecular weights of the FLO-H filtrate reducer and the VR-H viscosity reducer were 5.45 × 10^5^ g/mol and 8.51 × 10^3^ g/mol, respectively, and all of them showed good high-temperature resistance. The API filtration loss of the bentonite-base slurry with 3.0 wt% FLO-H was only 6.2 mL, which indicated that FLO-H had a prominent reduction in filtration loss after aging at high temperature. When the dosage of VR-H was 1.0 wt%, the plastic viscosity of the water-based drilling fluid after aging at 200 °C decreased from 71 mPa·s to 55 mPa·s, which provided excellent dispersion and dilution. The high-temperature and high-density water-based drilling fluid containing the FLO-H filtrate reducer and the VR-H viscosity reducer had good suspension stability and low filtration performance at the high temperature of 200 °C, which can meet the requirements of high-temperature deep well drilling.

## 1. Introduction

With the development of the petroleum industry and the consumption of shallow oil resources, oil and gas exploration has been gradually forced to transit from shallow reservoirs to deep strata, which creates a technical challenge for deep well drilling operations [1,2,3]. The drilling fluid in the drilling process of oil and gas wells can be seen as the equivalent of the blood in the human body, and high-temperature and high-density drilling fluid plays an important role in the whole deep well drilling operation. High-temperature and high-density drilling fluid has a high solid phase content and little free water content, so the rheological control means of this drilling fluid are limited and difficult [4]. Clay particles can form strong structures in drilling fluids, which can seriously affect their viscosity, especially in deep wells. Slurry bentonite has strong hydration in high temperature environments, which eventually leads to the formation of excessive grid structure in high-density drilling fluids [5,6]. In addition, the intrusion of drilling cuttings in the harsh downhole environment and their strong dispersion in drilling fluid lead to the increase in the solid phase content of high-density water-based drilling fluid, which can further cause the obvious thickening effect on high-density drilling fluid at high temperature [7,8]. With respect to the thickening effect at high temperature, it can only be addressed by strengthening the removal of the solid phase during the drilling process and adding a continuous phase (water), but the stability of the drilling fluid is maintained by the further addition of additives, contributing to the rising cost of drilling [9]. Therefore, it is necessary to strictly control the rheological properties of drilling fluid to avoid the increase in equivalent circulating density (ECD) and the phenomenon of pressure leakage in the drilling process. It is also important to ensure that the drilling fluid can avoid surge pressure caused by tripping, which is conducive to wellbore stability [10]. In addition to improving the rheological properties of high-temperature and high-density drilling fluids, it is also important to control their filtration loss, which contributes significantly to wellbore stability and reservoir protection. Due to the increased thermal movement of molecules caused by temperature rises, the polymer additives of drilling fluid are prone to degradation and cross-linking, and the desorption between it and clay occurs easily, which affects the protective colloid ability of the polymer and makes the clay particles more dispersed [11]. Macroscopically, the increased filtration loss of the drilling fluid results in a mud cake with the characteristics of virtual thickness, no toughness and high permeability [12].

At present, the filtration loss and rheological properties of high-temperature and high-density water-based drilling fluid are difficult to control due to a defined high temperature and high solid phase content. Hence, the research objectives of petroleum industry experts and scholars are to develop high-performance filtrate reducers and viscosity reducers. It is required that the filtrate reducer molecules have strong adsorption capacity on the clay surface and that they be less affected by high temperature [13]. Factually, the hydration and adsorption groups in the filtrate reducer molecules can maintain their properties to reduce the fluid loss on the basis of chelation adsorption and hydrogen bonding adsorption. The viscosity reducer, a specific polymer, is a widely used additive for regulating rheology, and it is required to have high temperature resistance and avoid the presence of ether groups and ester groups. The molecular backbone and the backbone-hydrophilic group connection bonds use the “C-C” bond, “C-N” bond, “C-Si” bond, and “C-S” bond to improve the thermal stability of additives in drilling fluid [14]. The viscosity reducer embedded with strong hydrophilic ionic groups can reduce the dehydration effect by forming a complex with the polymer, which breaks up the bridge structure between clay mineral particles and the polymer [15]. The cross-linking between the macromolecular polymers can be clearly restrained due to the formed complex, which hinders the ability of the polymer drilling fluid to form the space grid structure, and thus reduces the characteristic viscosity of the macromolecular polymer.

Nowadays, the environment-friendly water-based drilling fluid system developed for the petroleum development industry cannot successfully withstand temperatures up to 180 °C, and its density is generally lower than 1.8 g/cm^3^, which cannot well meet the drilling requirements of high-temperature deep wells and ultra-deep wells [16,17]. The multipolymers with a temperature resistance of 180~200 °C and outstanding salt resistance are synthesized using functional vinyl monomers such as AM, AMPS, AA, and NVP. However, their biodegradability is poor, and the remaining reactive monomer in the product may have a certain toxicity [18,19]. In addition, part of the additives that are applied to water-based drilling fluid, such as sulfonic acid polymer and natural polymer modified sulfonic acid monomer, belong to the environment-friendly materials, and they rely on sulfonic acid groups for stability at high temperatures. However, the sulfonic acid group can prevent microorganisms from entering the active part of the polymer and restrict the movement of the polymer chain, which reduces its biodegradation activity [20]. The raw materials used as the main components of the additives in drilling fluid system, such as cellulose, starch and chitin, contain a large number of ether groups, thus reducing their thermal stability [21]. Natural plant polymers such as lignin and humic acid have good environmental friendliness, and they have been mainly modified with sulfonic acid in the existing literature [22]. However, there are few studies on the modification of non-sulfonated plant polymers. Natural plant compound molecules such as humic acid and lignin contain a large number of active hydroxyl groups, which can be used as adsorption groups and grafting points to react with vinyl monomers. At the same time, the molecule contains multiple benzene rings, thus giving it better thermal stability.

In this paper, the polycondensation reaction of humic acid and lignin under the action of formaldehyde was firstly performed to increase their molecular weight. Then, the polymer prepared above was grafted with a water-soluble multifunctional polymer monomer to prepare the high temperature FLO-H filtrate reducer and the high temperature VR-H viscosity reducer. Finally, the prepared additives were used as the key treatment agents to form the high-temperature and high-density water-based drilling fluid system based on non-sulfonated plant polymers.

## 2. Experimental Section

### 2.1. Materials

Formaldehyde, sodium methoxide, sodium hydroxide (NaOH), potassium chloride (KCl), sodium carbonate (Na_2_CO_3_), monoethanolamine (MEA), acrylic acid (AA), acrylamide (AM), N-vinylpyrrolidone (NVP), N-(3-triethoxymethiloxanyl) propyl acrylamide (APTS), benzoyl peroxide (BPO) were all of analytical purity and supplied by Sinopharm Chemical Reagent Co., Ltd.,(Shanghai, China), wherein NaOH and Na_2_CO_3_ already work together to control the pH value of the water-based drilling fluid, and KCl works as a conventional inhibitor and antioxidant. Technical grade humic acid and lignin were purchased from BASF Co., Ltd., (Shanghai, China). Nano blocking agent (code HISEAL), and bio-oil-modified lubricant (code LUHE-H) as the main components for the construction of water-based drilling fluid were purchased from China Nanhai Magcobar Mud Co., Ltd., (Shenzhen, China). Sodium bentonite, polymer emulsion high temperature stabilizer (code MG-H) and barite were obtained from Jingzhou Jiahua Tech. Co., Ltd., (Jingzhou, China). All of the chemicals were used as received without further treatment.

### 2.2. Preparation of FLO-H Filtrate Reducer

The filtrate reducer was prepared using a two-step procedure consisting of a polycondensation reaction and free radical polymerization in aqueous solution; the synthesis route of FLO-H is shown in Figure 1. An amount of 200 g deionized water, 80 g humic acid and 20 g lignin were firstly added into a four-mouth flask equipped with a stirrer, reflux condenser, thermometer and nitrogen inlet and outlet pipe, and the solution was stirred for 2 h at room temperature to make the reaction materials fully and evenly mixed. After that, 5 g formaldehyde solution was put into the above four-mouth flask, and the pH value of the whole solution was adjusted to 8 with NaOH particles, followed by keeping the reaction temperature at 70 °C and stirring for 3 h. The above operation can complete the polycondensation reaction of the reaction raw materials. Then, 10 g AM, 10 g NVP, 10 g AA and 2 g APTS were added to the above solution with continued stirring for 1 h, which ensures that the raw materials are evenly mixed. After passing N_2_ through for 30 min, 0.5 g BPO, a common sulfur-free initiator, was added to the mixture to catalyze the polymerization reaction, followed by keeping the reaction temperature at 80 °C and stirring for 6 h. The above operation can complete the free radical polymerization reaction of the reaction raw materials. Finally, the final product was obtained by drying and crushing and named the FLO-H filtrate reducer.

### 2.3. Preparation of VR-H Viscosity Reducer

The viscosity reducer was prepared using a three-step procedure consisting of a polycondensation reaction, free radical polymerization in aqueous solution and an amidation reaction. The synthesis route of VR-H is shown in Figure 2. The polycondensation reaction was firstly completed with 20 g humic acid, 80 g lignin and 5 g formaldehyde solution as raw materials, and other reaction conditions were consistent with the polycondensation reaction conditions in the preparation of the filtrate reducer. Then, 10 g AM, 10 g AA and 2 g APTS were added to the above reaction flask with continued stirring for 1 h. After passing N_2_ through for 30 min, 0.8 g BPO was added into the mixture to catalyze the polymerization reaction, followed by keeping the reaction temperature at 70 °C and stirring for 3 h. The above operation can complete the free radical polymerization reaction. Finally, after adding 10 g monoethanolamine (MEA) and 1 g sodium methanol (catalyst), the mixture was put into a vacuum oven at 150 °C for 6 h, which allows it to undergo the amidation reaction and drying. The final product was obtained by crushing and named the VR-H viscosity reducer.

### 2.4. Methods

#### 2.4.1. Characterization of the Prepared Samples

FT-IR spectra were measured with a WQF-520 Fourier Transform Infrared (FT-IR) spectrometer (Beijing Jingke Ruida Technology Co., Ltd., Beijing, China) by the KBr tablet method. The molecular structure of the samples was analyzed using an infrared spectral width in the range of 500~4000 cm^−1^. Gel Permeation Chromatography (GPC) was utilized to measure the molecular weight of the samples using an Alliance e2695 instrument (Waters, Waltham, MA, USA); the injection volume was 50 mL, the testing temperature was 30 °C, the flow rate was 1.0 mL/min, the standard sample was polystyrene and the solvent was tetrahydrofuran. The thermal stability analysis of samples was carried out under a nitrogen atmosphere with a heating rate of 5 °C/min using a Q500 series Thermogravimetric Analyzer (Shanghai Lerui Scientific Instrument Co., Ltd., Shanghai, China). At a certain temperature, the temperature resistance of samples was judged by the mass retention rate.

#### 2.4.2. Preparation of Bentonite-Base Slurry

Firstly, 0.8 g Na_2_CO_3_, 0.6 g NaOH, and 16 g sodium bentonite were added to 400 mL of the deionized water. At room temperature, the mixture was then stirred for 30 min at 10,000 rpm through a high-speed agitator. Finally, the mixture was static for 24 h in an airtight container, at which point it was a bentonite-base slurry.

#### 2.4.3. Performance Evaluation of FLO-H

The temperature resistance and reduction in filtration loss of the filtrate reducer are judged by rheological properties and the filtration loss [23]. The above bentonite-base slurry with filtrate reducer (FLO-H) was aged at 200 °C for 16 h. The rheological properties of the bentonite-base slurry with filtrate reducer before and after the aging process were measured at six specific shear rates, i.e., 600, 300, 200, 100, 6, and 3 rpm, using a ZNN-D6 size rotational viscometer (Qingdao Hengtaida Electromechanical Equipment Co., Ltd., Qingdao, China), which was applied to calculate apparent viscosity (AV), plastic viscosity (PV) and yield point (YP). The rheological properties were tested at a temperature of 50 °C. The calculation formulas are shown in formulas 1–3. The initial gel strength (Gel _10 s_) and the 10-min gel strength (Gel _10 min_) were recorded as half of the maximum dial reading at θ_3_ after standing undisturbed for 10 s and 10 min, respectively.

In addition, according to the American Petroleum Institute (API) standards, the API filtration loss (FL_API_) of the bentonite-base slurry with filtrate reducer was evaluated through the ZNS size of the mud loss apparatus (Qingdao Hengtaida Electromechanical Equipment Co., Ltd., Qingdao, China).
AV = θ_600_/2 (mPa·s)(1)
PV = θ_600_ − θ_300_ (mPa·s)(2)
YP = 0.511(θ_300_ − PV) (Pa)(3)

#### 2.4.4. Performance Evaluation of VR-H

The high-temperature and high-density water-based drilling fluid system with the viscosity reducer VR-H was aged at 200 °C for 16 h. The rheological properties of the drilling fluid before and after the aging process were measured to evaluate the viscosity-reducing effect of VR-H. This was tested at a temperature of 50 °C. Moreover, the effect of the viscosity reducer on the filtration loss of the drilling fluid was evaluated to further analyze its performance. The formula of the high-temperature and high-density water-based drilling fluid was: 2.5 wt% bentonite-base slurry + 0.3 wt% NaOH + filtrate reducer (FLO-H) + viscosity reducer (VR-H) + 2.0 wt% lubricant (LUHE-H) + 3.0 wt% high temperature stabilizer (MG-H) + 3.0 wt% blocking agent (HISEAL) + 7.0 wt% KCl + barite (ρ = 2.0 g/cm^3^).

## 3. Results and Discussion

### 3.1. Analysis of Molecular Properties and Filtration Loss of FLO-H

#### 3.1.1. Analysis of Infrared Spectrum and Molecular Weight of FLO-H

The FT-IR spectra of FLO-H are shown in Figure 3. As indicated, the spectrum absorption peaks at 1554 and 1457 cm^−1^ were the stretching vibration of benzene ring structures in plant polymer molecules including humic acid and lignin. The peak at 3490 cm^−1^ was the stretching vibration absorption peak of hydroxyl groups in plant polymer molecules and 2951 cm^−1^ belonged to the stretching vibration of C–H on the backbone of polymer molecules. The spectrum absorption peaks at 1390 and 1351 cm^−1^ were the stretching vibrations of C–N–C in the NVP unit. The strong absorption peak at 1682 cm^−1^ did correspond to the stretching vibration of amide groups in the AM unit, and the absorption peak at 1437 cm^−1^ did correspond to the stretching vibration of carboxyl groups in the AA unit. The peak at 1012 cm^−1^ was the absorption peak of –O–Si in the APTS unit. This indicated that humic acid, lignin and the multi-function monomer were successfully copolymerized. In addition, the molecular weight of FLO-H was measured by gel permeation chromatography (GPC). The GPC test results showed that the M_w_ of FLO-H was approximately 5.45 × 10^5^ g/mol.

#### 3.1.2. Thermogravimetric Analysis of FLO-H

A Thermogravimetric Analyzer was utilized to investigate the thermal stability of FLO-H. The thermal gravimetric curve displayed three stages of weight loss, which are shown in Figure 4. The first stage appeared in the range of 50∼330 °C, and the mass of FLO-H was lost due to the evaporation of intermolecular water and free water. The second stage took place in the temperature range 330∼440 °C with a loss of 49.8 mass (wt%), and was probably mainly due to the decomposition of the functional groups such as the amide, carboxyl and silica groups. The final stage occurred in the temperature range 440∼600 °C, and showed the main structure and pyrrolidone of FLO-H to be mostly destroyed. In general, the thermal decomposition of FLO-H began after 330 °C, which indicated that it had good thermal stability.

#### 3.1.3. Performance Evaluation of FLO-H

The above bentonite-base slurry with different amounts of filtrate reducer (FLO-H) was aged at 200 °C for 16 h. Its rheological properties and the API filtration loss (FL_API_) were measured, and the experimental results are shown in Figure 5.

Figure 5 presents the typical rheological and filtrate data of the bentonite-base slurry, including PV and YP. With increasing amounts of FLO-H, the PV and YP of the bentonite-base slurry with FLO-H gradually increased, and the filtration loss gradually decreased. When the amount of FLO-H was 3.0 wt%, the PV and YP of the aged slurry were as high as 17 mPa·s and 12 Pa, respectively, and the API filtration loss was only 6.2 mL. These results indicated that the molecular weight of FLO-H can be effectively increased by condensation between humic acid and lignin, and it was applied to the bentonite-base slurry, which shows excellent performance as it increases viscosity and shear strength after aging at high temperature, and effectively reduces the filtration loss. The analysis of the filtration control mechanism of FLO-H is shown in Figure 6. Humic acid and lignin were mainly composed of aromatic rings, aliphatic groups, an adsorption group and a hydration group, and they were based on carbon and aromatic rings as the skeleton, exhibiting good resistance to high temperature. Humic acid and lignin in FLO-H molecules contained a large number of hydroxyl and carboxyl groups, forming coordination bonds between the adsorption group and aluminum ions on the broken edges of the clay particles. Multiple carboxyl groups made the hydration film on the surface of clay particles thicker under the action of hydration, increasing the absolute value of the Zeta potential of the clay particles, which can prevent clay particles from colliding with each other. In addition, the side chain of FLO-H with AM and NVP contained a rigid pyrrolidone and an amide group, which can enhance the rigidity of the molecular chain and inhibit the degradation of the molecular backbone and side groups at high temperature, and thus ensure its stability at high temperature [24]. The siloxane groups of FLO-H were hydrolyzed in aqueous solution to generate Si–OH bonds, which can react with Si–OH bonds on the hydrated clay surface to form strong chemical adsorption. The chemical adsorption was very stable at high temperature, which can further prevent clay particles from colliding with each other [25]. FLO-H can effectively adsorb on the surface of the bentonite to form a spatial network structure. This was conducive to the formation of a compact filter cake, thus greatly reducing the filtration loss of the slurry.

### 3.2. Analysis of the Molecular Properties of the VR-H Viscosity Reducer 

#### 3.2.1. Analysis of Infrared Spectrum and Molecular Weight of VR-H

The FT-IR spectra of VR-H is shown in Figure 7. As indicated, the spectrum absorption peaks at 1539 and 1448 cm^−1^ were the stretching vibrations of benzene ring structures in plant polymer molecules including humic acid and lignin. The peak at 3481 cm^−1^ was the stretching vibration absorption peak of hydroxyl groups and 2939 cm^−1^ belonged to the stretching vibration of C–H on the backbone of polymer molecules. The strong absorption peak at 1658 cm^−1^ did correspond to the stretching vibration of amide groups, and 1050 cm^−1^ was the absorption peak of –O–Si in the APTS unit. The obvious characteristic peak, an absorption peak of the carboxyl group, was not found in the shift range of 1440~1350 cm^−1^ in the FT-IR spectra, which indicated that AA had already undergone an amidation reaction with monoethanolamine. This indicated that humic acid, lignin and the multi-function monomer were successfully copolymerized. In addition, the molecular weight of VR-H was measured by gel permeation chromatography (GPC). The GPC test results showed that the M_w_ of VR-H was approximately 8.51 × 10^3^ g/mol.

#### 3.2.2. Thermogravimetric Analysis of VR-H

A thermogravimetric analyzer was utilized to investigate the thermal stability of VR-H. The thermal gravimetric curve is shown in Figure 8. The mass retention rate of VR-H at 300 °C was still as high as 89.6 wt%. The mechanism of its mass loss was similar to that of the FLO-H filtrate reducer, which was mainly caused by the volatilization of free water and intermolecular water. Therefore, the main functional groups of VR-H were relatively stable at temperatures below 300 °C.

#### 3.2.3. Performance Evaluation of VR-H

The dispersion and dilution ability of VR-H was evaluated in the above drilling fluid containing 3 wt% FLO-H. The experimental results are shown in Figure 9.

As indicated in Figure 9, the drilling fluid without VR-H presented serious high temperature thickening after aging, and its rheological property was not conducive to the drilling operation. However, with the addition of VR-H, the rheological property of the high-density water-based drilling fluid was gradually improved. VR-H was conducive to reducing filtration loss. When the VR-H addition was 1 wt%, the rheological properties and filtration performance of the high-density water-based drilling fluid were the best. Therefore, the optimal amount of VR-H was 1 wt%. VR-H was a low-molecular-weight polymer that provided excellent dispersion and dilution in high-density drilling fluid, and analysis of the viscosity reduction mechanism of VR-H is shown in Figure 10. Humic acid and lignin have a carbon and aromatic ring-based skeleton, exhibiting good resistance to high temperature, viscosity reduction and poor water solubility. The viscosity reducer VR-H contained a large number of amide groups, hydroxyl groups and amine groups under the action of AM, MEA and AA modification, increasing its water solubility and temperature resistance. The VR-H viscosity reducer contained a large number of hydroxyl, silicon hydroxyl and amide groups adsorbed on the surface of solid particles (clay particles and barite), which can effectively increase the electrostatic repulsion and the thickness of the hydration film on the surface of solid particles. It can break up the end-surface and end-end frame structure formed between the clay particles in the drilling fluid, thus affording dilution and viscosity reduction. The VR-H viscosity reducer also formed a polymer complex with polyacrylamide in the water-based drilling fluid system, which can reduce the adsorption bridging effect of polyacrylamide on clay particles and the interaction between hydrolyzed polyacrylamide molecules, so that the structural viscosity and liquid viscosity of the drilling fluid are decreased. Therefore, VR-H can keep the solid particles in drilling fluid well dispersed, prevent the passivation and aggregation of clay at high temperature, reduce the friction between solid particles, and thus reduce the viscosity of the drilling fluid [26].

### 3.3. Performance Evaluation of the High-Temperature and High-Density Water-Based Drilling Fluid System

#### 3.3.1. Effect of Aging Times on the Drilling Fluid

The FLO-H filtrate reducer and VR-H viscosity reducer were used as the key additives to form a high-temperature and high-density water-based drilling fluid. The formula of the drilling fluid was: 2.5 wt% bentonite-base slurry + 0.3 wt% NaOH + 3.0 wt% filtrate reducer (FLO-H) + 1.0 wt% viscosity reducer (VR-H) + 2.0 wt% lubricant (LUHE-H) + 3.0 wt% high temperature stabilizer (MG-H) + 3.0 wt% blocking agent (HISEAL) + 7.0 wt% KCl + barite (ρ = 2.0 g/cm^3^). The rheological and filtration properties of the drilling fluid were evaluated after different aging times, and the experimental results are shown in Figure 11.

As indicated in Figure 11, with the extension of aging time, the plastic viscosity, yield point, the initial gel strength (Gel _10 s_) and 10-min gel strength (Gel _10 min_) of the drilling fluid decreased slowly, and its filtration loss was less than 5 mL. As aging time was increased from 16 h to 48 h, the drilling fluid still had good gel strength. This indicated that the drilling fluid had good ability to suspend barite and carry cutting. Therefore, there was no barite settling phenomenon. Obviously, the drilling fluid system had good suspension stability and low filtration performance at the high temperature of 200 °C, which can meet the requirements of high-temperature deep well drilling.

#### 3.3.2. Effect of Pollutants on the Drilling Fluid

During the drilling process of high-temperature deep wells, the drilling fluid was easily polluted by the hydratable cations Na^+^ and Ca2^+^, which can lead to the deterioration of the rheological and filtration performance of the drilling fluid. The rheological and filtration performance of the drilling fluid with pollutants was evaluated after aging at 200 °C for 16 h, and the experimental results are shown in Figure 12.

As indicated in Figure 12, the plastic viscosity, yield point and filtration loss of the drilling fluid with pollutants were still reasonable; the resistance to the hydratable cuttings was 10 wt%, the salt resistance was 10 wt%, and the calcium resistance was 1.0 wt%, which indicated that the drilling fluid had excellent high temperature stability and resistance to pollutants. Based on the above analysis, this high-temperature and high-density water-based drilling fluid can meet the technical requirements of high-temperature deep wells.

## 4. Conclusions

Environment-friendly water-based drilling fluids cannot successfully withstand temperatures up to 180 °C. In addition, most high-temperature resistant additives with sulfonic acid groups that have been successfully applied to water-based drilling fluid are not good for environmental protection. A non-sulfonated filtrate reducer and a viscosity reducer with resistance to high temperature were prepared to solve the above technical problems. According to a polycondensation reaction, free radical polymerization in aqueous solution and an amidation reaction, the non-sulfonated filtrate reducer and viscosity reducer with resistance to high temperature were prepared by using humic acid, lignin and a multifunctional monomer as raw materials. The molecular weights of the FLO-H filtrate reducer and the VR-H viscosity reducer were 5.45 × 10^5^ g/mol and 8.51 × 10^3^ g/mol, respectively. The main functional groups of FLO-H and VR-H were relatively stable at temperatures below 300 °C, thus exhibiting good high-temperature resistance. The API filtration loss of the bentonite-base slurry with 3.0 wt% FLO-H was only 6.2 mL, which indicated that FLO-H had a prominent reduction in filtration loss after aging at high temperature. The drilling fluid with 1.0 wt% VR-H showed good rheological properties, and there was no barite settling phenomenon. The high-temperature and high-density water-based drilling fluid containing the FLO-H filtrate reducer and the VR-H viscosity reducer had excellent high temperature stability and resistance to pollutants, which meets the technical requirements of high-temperature deep wells.

## Figures and Tables

**Figure 1 polymers-14-04433-f001:**
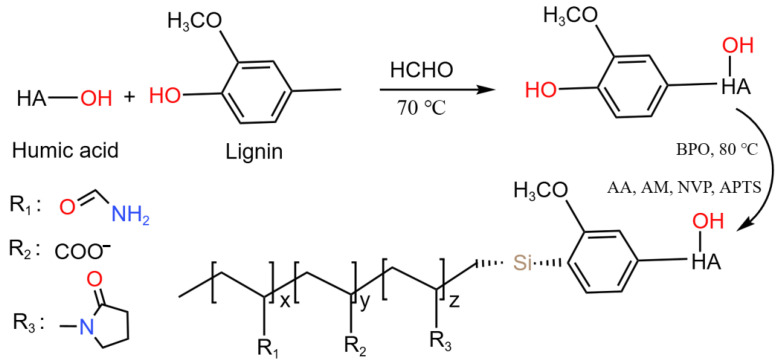
Synthesis route of FLO-H.

**Figure 2 polymers-14-04433-f002:**
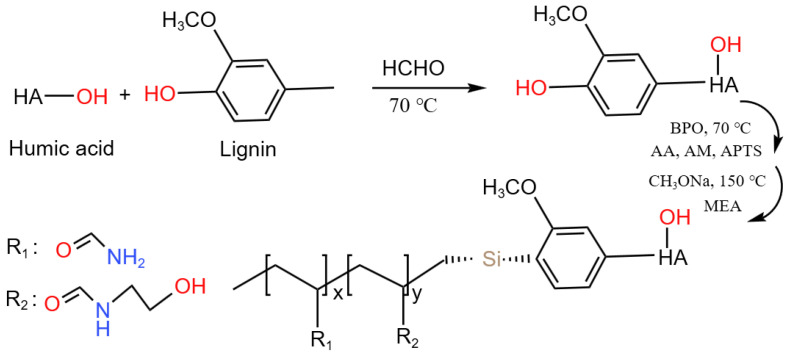
Synthesis route of VR-H.

**Figure 3 polymers-14-04433-f003:**
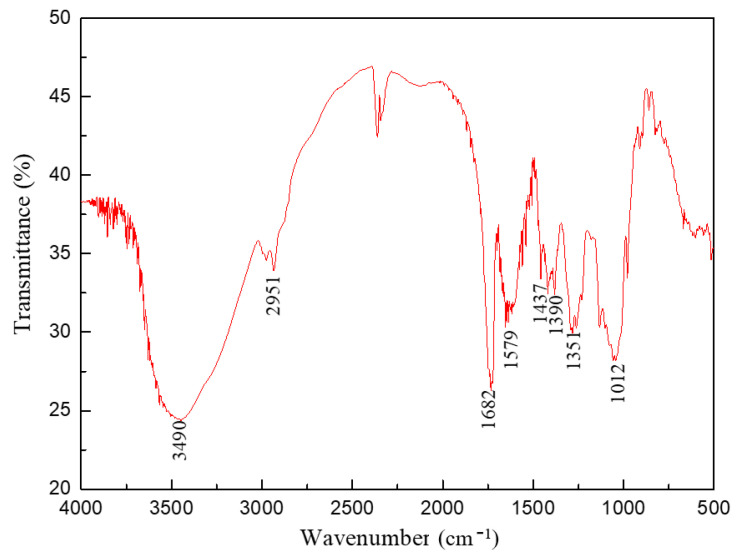
The FT-IR spectra of FLO-H.

**Figure 4 polymers-14-04433-f004:**
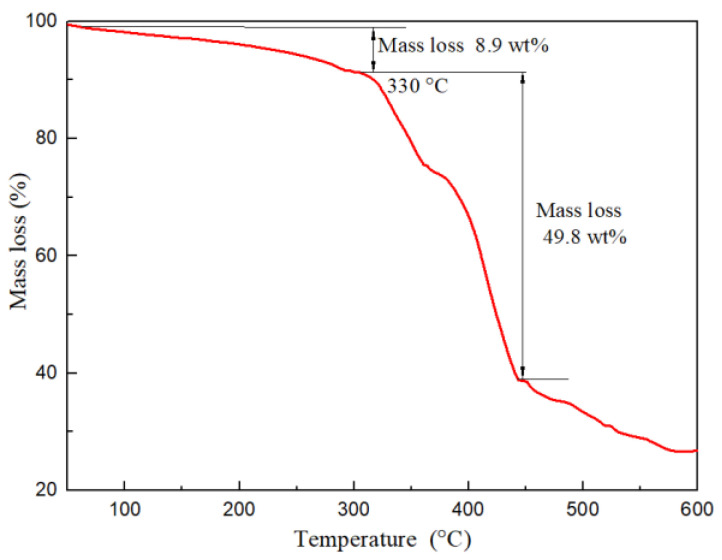
TG curve of FLO-H.

**Figure 5 polymers-14-04433-f005:**
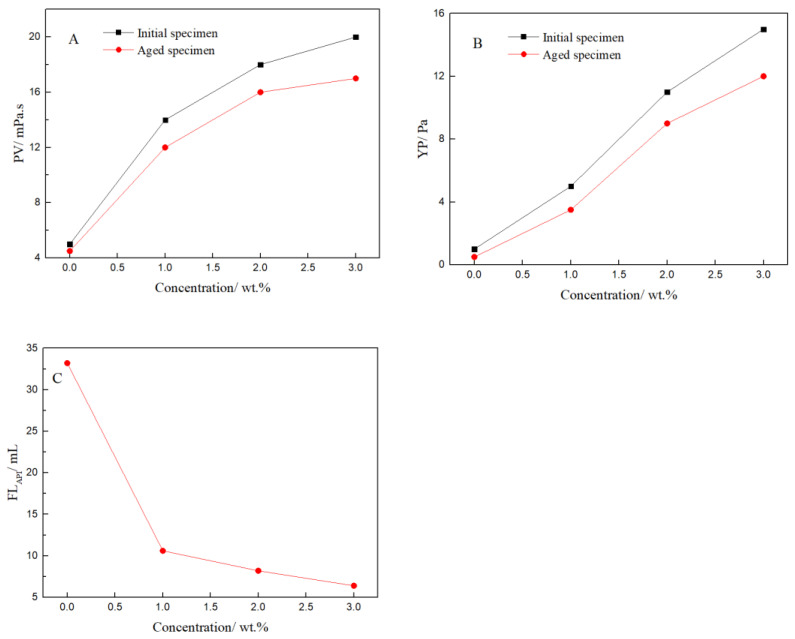
Effect of FLO-H concentration on rheological and filtrate properties of the bentonite-base slurry: (**A**) Plastic viscosity (PV), (**B**) Yield point (YP), (**C**) FL_API_ of the bentonite-base slurry with FLO-H after aging.

**Figure 6 polymers-14-04433-f006:**
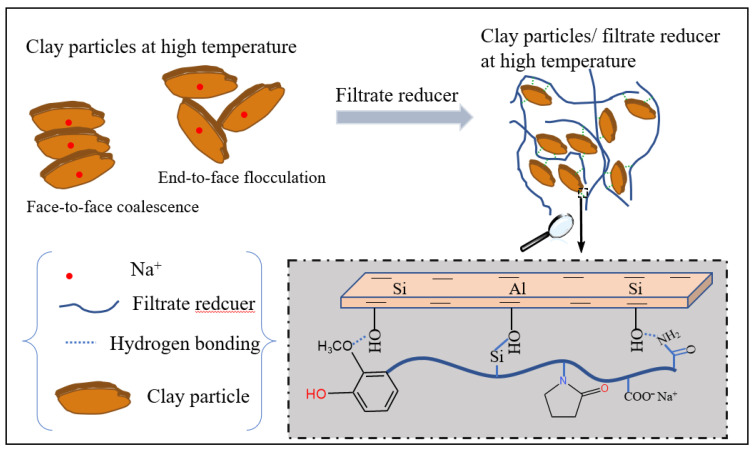
Analysis of the filtration control mechanism of FLO-H.

**Figure 7 polymers-14-04433-f007:**
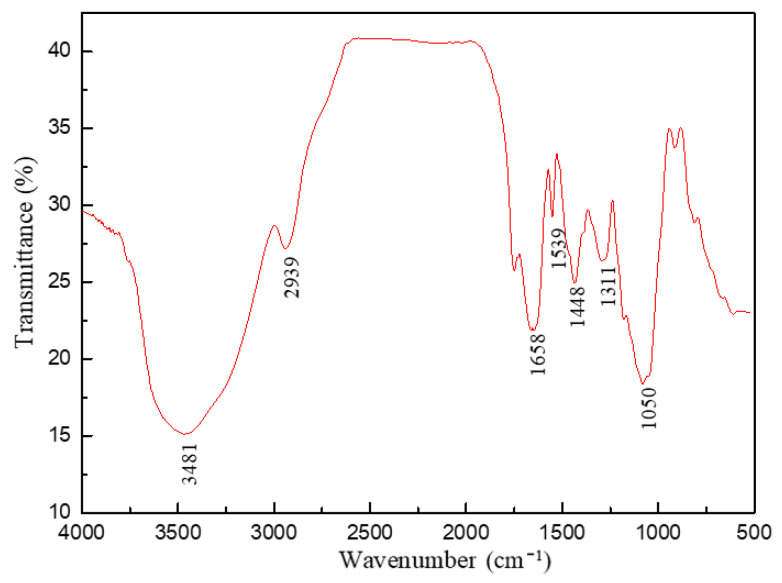
The FT-IR spectra of VR-H.

**Figure 8 polymers-14-04433-f008:**
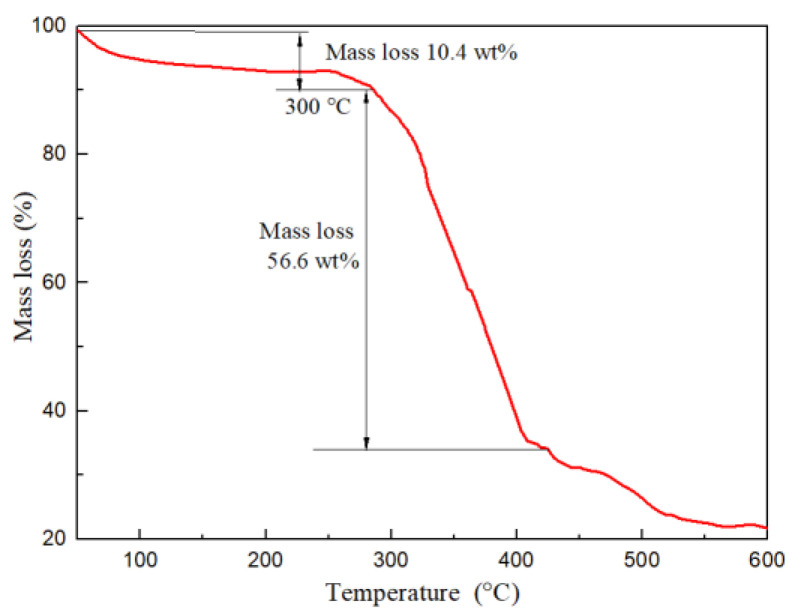
TG curve of VR-H.

**Figure 9 polymers-14-04433-f009:**
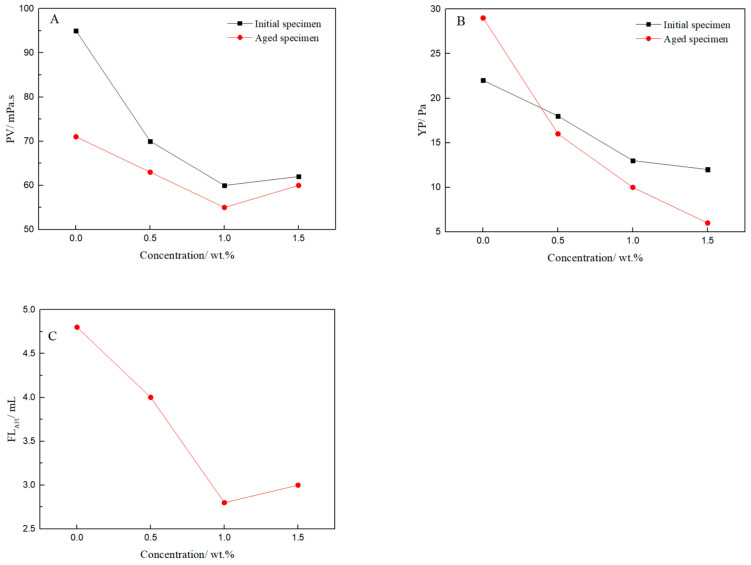
Effect of VR-H concentration on rheological and filtrate properties of the drilling fluid: (**A**) Plastic viscosity (PV), (**B**) Yield point (YP), (**C**) FL_API_ of the drilling fluid after aging.

**Figure 10 polymers-14-04433-f010:**
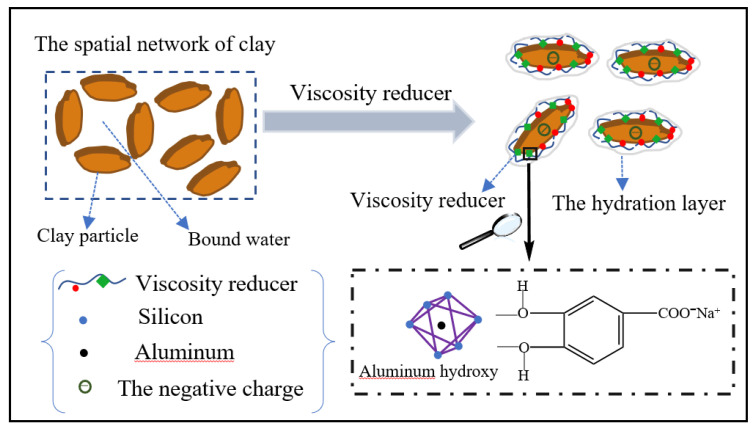
Analysis of the viscosity reduction mechanism of VR-H.

**Figure 11 polymers-14-04433-f011:**
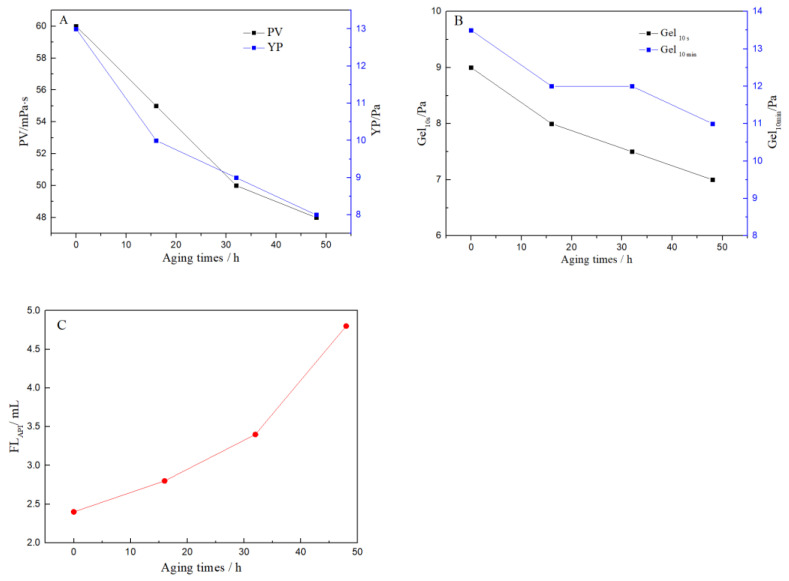
Effect of aging times on the drilling fluid: (**A**) Plastic viscosity (PV) and Yield point (YP), (**B**) The initial gel strength (Gel _10 s_) and 10-min gel strength (Gel _10 min_), (**C**) FL_API_ of the drilling fluid.

**Figure 12 polymers-14-04433-f012:**
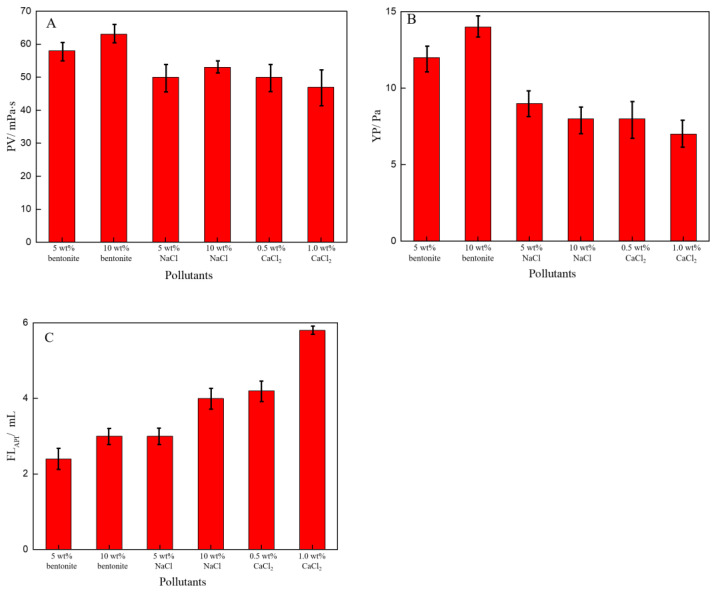
Effect of pollutants on the drilling fluid: (**A**) Plastic viscosity (PV), (**B**)Yield point (YP), (**C**) FL_API_ of the drilling fluid. Data were shown as mean ± s. e. m. (n 3, replicates).

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
