# Peer review of "Study of a High-Temperature and High-Density Water-Based Drilling Fluid System Based on Non-sulfonated Plant Polymers"

_polymers, 2022, doi:10.3390/polym14204433_

Round 1

Reviewer 1 Report

High temperature is detrimental to clay/polymer complex in conventional water-based mud. Naturally occurring materials like cellulose and starch contain ether groups that degrade at higher temperatures. Additionally, the presence of sulfonic groups in thermally tolerant polymers can be a problem in terms of biodegradability. This manuscript addresses this issue by synthesizing non-sulfonated plant polymers and their applications as filtrate reducers and viscosity reducers in high-density water-based drilling mud at high temperatures (200℃).

A few observations after reading the manuscript:

·       There are a few grammatical issues that can be corrected with a thorough reading. For example, in section 1, the word “…choose” should be “chosen”.

·       In section 2.4.2, the addition of Na2CO3 to deionized water is unclear as they are basically used to condition tap water for instance. NaOH has already been used as a pH regulator.

·       In section 3.1.3., the mechanism of the increasing zeta potential of clay particles with FLO-H can be substantiated with some zeta potential measurements, if possible.

·       In section 3.3.1., the claim of no barite settling slightly contrasts with the given PV and YP values. The inclusion of gel strength data (10s/10min) and/or sag index is recommended for further clarification. 

Author Response

Dear reviewer

Thank you for your valuable amendments and guidance.

  • There are a few grammatical issues that can be corrected with a thorough reading. For example, in section 1, the word “…choose” should be “chosen”.
  • Answer:a few grammatical issues had been corrected with a thorough reading.

  • In section 2.4.2, the addition of Na2COto deionized water is unclear as they are basically used to condition tap water for instance. NaOH has already been used as a pH regulator.
  • Answer:NaOH is used as a pH regulator. The role of Na2CO3 is to adjust the performance of drilling fluid. The addition of Na2CO3 is to improve the calcium resistance of drilling fluid. And it can be used as pH buffer to help polymer show better performance in drilling fluid.

  • In section 3.1.3., the mechanism of the increasing zeta potential of clay particles with FLO-H can be substantiated with some zeta potential measurements, if possible.
  • Answer: Polymer, as a filtrate reducer for drilling fluid, has long been studied. It contains a large number of adsorption groups, carboxyl groups and sulfonic acid groups. The adsorption group can adsorb onto the surface of the clay particles. In addition, carboxyl and sulfonic acid groups can enhance the Zeta potential (absolute value) of clay particles, which can prevent clay particles from colliding with each other. Scholars have verified this view many times. The mechanism of the increasing zeta potential of clay particles with filtrate reducer will be systematically analyzed in the next study.

  • In section 3.3.1., the claim of no barite settling slightly contrastswith the given PV and YP values.The inclusion of gel strength data (10s/10min) and/or sag index is recommended for further clarification
  • Answer:In section 3.3.1., gel strength data (10s/10min) was added to Figure 11, and it was analysed to explain the claim of no barite settling.

Reviewer 2 Report

In this work, the authors studied high temperature and high density water-based drilling fluid system based on Non-sulfonated plant polymer. The topic is interesting and consists of “polymers”. However, the authors need to answer the following questions before we accept this work.

1.       Line 190: Can the authors help explain what are apparent viscosity, plastic viscosity, and yield point? What are the physical means of that? What are the potential applications of that?  

2.       Figure 5. C: What are the possible reasons of the dramatically drop of FL when concentration increase from 0 to 1.0%?

3.       Figure 5 in general, I suggest replotting figure 5.  Increasing the font size, the symbol size, and insert sub-ticks if possible.

4.        Figure 6 and figure 10, the style should be consistent.

5.       Figure 12.  How many times did you repeat the tests for this plot? Usually, it is necessary to show the error bar of the results. Otherwise, we cannot claim whether the results have a statistical difference.

6.       Conclusions:  Please clearly point out the difference between this work (innovation point)?

Author Response

Dear reviewer

Thank you for your valuable amendments and guidance.

  • 1.Line 190: Can the authors help explain what are apparent viscosity, plastic viscosity, and yield point? What are the physical means of that? What are the potential applications of that?
  • Answer: The internal friction between solid particles and liquid phase is called plastic viscosity when the destruction and recovery of the grid structure in drilling fluid are in dynamic equilibrium. Cuttings are efficiently carried in drilling fluid with a certain plastic viscosity. However, if the plastic viscosity of drilling fluid is too high, the fluidity of the drilling fluid will become poor, which is not conducive to the removal of cuttings on the drilling platform during the drilling process.    The interaction force between clay particles and polymer molecules is called the yield point. The yield point represents the ability of the drilling fluid to suspend cuttings. The larger the yield point, the stronger the rock carrying effect of drilling fluid.    For drilling fluid, the ratio of shear stress to shear rate is defined as the apparent viscosity of drilling fluid. The value of apparent viscosity is the sum of plastic viscosity and yield point. The drilling fluid with a certain apparent viscosity can effectively carry cuttings. However, if the apparent viscosity of drilling fluid is too high, the fluidity of the drilling fluid will become poor, which is not conducive to wellbore stability during drilling and tripping.

  • Figure 5. C: What are the possible reasons of the dramatically drop of FL when concentration increase from 0 to 1.0%?
  • Answer:  The dispersive clay particles in the bentonite-base slurry are easy to accumulate at high temperature, which leads to mud cake with high permeability formed by drilling fluid under certain pressure difference.     The clay particles in the bentonite-base slurry containing 1% filtrate reducer are not easy to gather at high temperatures, because the molecules of the filtrate reducer can stably adsorb on the surface of the clay particles to prevent the aggregation of clay particles. Highly dispersed clay particles and filtrate reducer can form a stable grid structure, which is conducive to improving the compactness of mud cake. The hydration group of filtrate reducer molecules can effectively block the pores of mud cake and reduce the permeability of mud cake. In addition, the filtrate reducer improves the viscosity of bentonite-base slurry and reduces the mobility of free water. Therefore, FL can dramatically drop when concentration increase from 0 to 1.0%.

  • Figure 5 in general, I suggest replotting figure 5.  Increasing the font size, the symbol size, and insert sub-ticks if possible.
  • Answer:  I have replotted figure 5 and increased the font size, the symbol size.

  • Figure 6 and figure 10, the style should be consistent.
  • Answer:  The style of Figure 6 and Figure 10 I drew is consistent. In Figures 6 and 10, the state of clay without treatment agent is at the upper left, the state of clay with treatment agent is at the upper right, the micrograph of clay with treatment agent is at the lower right, and the symbol description is at the lower left.
  • Figure 12. How many times? Usually, it is necessary to show the error bar of the results. Otherwise, we cannot claim whether the results have a statistical difference.
  • Answer:  I had repeated 3 tests for this plot, and Figure 12 had been modified.
  • Conclusions:Please clearly point out the difference between this work (innovation point)?
  • Answer:  The innovation of this article has been rewritten in the conclusion.